# Effects of Propoxur Exposure on Insecticidal Susceptibility and Developmental Traits in *Culex pipiens quinquefasciatus*

**DOI:** 10.3390/insects10090288

**Published:** 2019-09-07

**Authors:** Xiaolei Zhang, Samuel Karungu, Quanxin Cai, Zhiming Yuan, Xiaomin Hu

**Affiliations:** Key Laboratory of Special Pathogens and Biosafety, Wuhan Institute of Virology, Chinese Academy of Sciences, Wuhan 430071, China (X.Z.) (S.K.) (Q.C.)

**Keywords:** *Culex pipiens quinquefasciatus*, propoxur, sublethal and lethal exposure, resistance, developmental traits

## Abstract

Propoxur-sel strains of *Culex pipiens quinquefasciatus* were derived from a lab-bred strain following 16 generations of propoxur exposure under sublethal concentrations of LC_25_ (lethal concentration of 25%) and LC_50_ (lethal concentration of 50%), respectively. This resulted in resistance development in F16 with ratios of 8.8× and 6.3×, respectively, compared with F0. The fecundity, longevity, sex ratio (F/M), and hatchability of the propoxur-exposed *Cx. quinquefasciatus* adult survivors and their offspring were decreased, with no effect on the emergence ratio and pupa survival rate. In addition, the intrinsic rates of increase (r), the net reproduction (R_0_), and the finite rate of increase (λ) of the *Cx. quinquefasciatus* offspring generations were also decreased significantly compared to F0. Correspondingly, the mean generation time (T) and the population double time (DT) in propoxur-sels were increased. Enhanced activities of cytochrome P450 monooxygenase and esterase were also observed in propoxur-sels, indicating that a detoxification mechanism might be responsible for resistance development in *Cx. quinquefasciatus*. Except for the three genes *cyp4d42v1*, *cyp4c52v1*, and *cyp6aa9* which displayed a coincidence in some degree in different treatments, induction by different doses of propoxur and constitutive expression in different generations of propoxur-sel strains resulted in an inconsistent identification of the P450 genes probably related with resistance.

## 1. Introduction

*Culex pipiens quinquefasciatus* is a common nuisance biting insect around the world. It is also a primary vector of many human ailments (e.g., filariasis, encephalitis, West Nile and Zika virus—infecting diseases) that pose a major global health threat [1,2,3,4,5]. Vector control is an effective preventive approach against mosquito biting and major vector-borne diseases, since this intervention could reduce human–vector contact and vector survival; thus, the transmission of these diseases can be suppressed and even halted [6,7,8]. Therefore, insecticides have been massively used and expanded for mosquito control [7,9].

The concentrations of insecticide spread into the environment are usually varied [10] and switch from lethal (when initially applied) to sublethal (with the degradation over time). On one hand, the breeding sites of the immature stages of mosquitos (larvae and pupae) are always aquatic environments, e.g., puddles, ditches, and rice fields [11], which could be contaminated by agrochemical pesticides and pose an opportunity for chronic and/or subchronic exposure [12]. On the other hand, due to limited exposure time and spatial range and to delayed onset of killing, the adult mosquito populations could also experience exposure to sublethal concentrations of insecticides [13]. Thus, mosquitoes are likely to be exposed to sublethal doses during the control application of pesticides, which could affect mosquito populations directly or indirectly [8,13,14]. Previous studies have shown that continued exposure to sublethal doses can lead to the emergence/development of insecticide-resistant populations, as well as sublethal effects on insect physiology and behavior [8,15,16].

The mechanism of resistance could be complicated and caused by a variety of genetic modifications related with metabolic detoxification and target site insensitivity [17]. A correlation has also been established in previous studies between the degree of resistance and enzyme activity expressed in mosquito, including esterase, glutathione S-transferase (GST), and cytochrome P450 monooxygenase [7]. In addition, structural modification of acetylcholinesterase (AChE), encoded by the ace-1 gene, which is a key enzyme promoting nerve signal transmission in both vertebrates and invertebrates, could make the mosquito insensitive to organophosphate and carbamate pesticides [18]. Moreover, agrochemical residues may affect the microbiome of mosquitoes and are commonly associated with some fitness traits of the dipteral insects [19,20].

Carbamate insecticides have been recommended for mosquito control due to their lower dermal toxicity and less unfavorable neurotoxic properties than organochlorine insecticides [21]. Propoxur, a widely used broad-spectrum carbamate insecticide, was introduced as an insecticide in 1959 [22,23]. Because of its fast killing and long residual effect, it has been widely used in hygienic, store house, and agricultural insect pest control with tremendous success. For instance, it was developed as cockroach poison and has been used against mosquitoes within long-lasting insecticidal nets (LLINs) and in- and outdoor residual spraying (ORS and IRS) programs. Consequently, it has been frequently detected in agricultural and aquatic environments due to its massive use [24]. However, the influence of short- and long-term exposure to sublethal propoxur on multiple traits associated with the life cycle of these insects remains unclear.

In this study, the susceptibility shift of *Cx. quinquefasciatus* to propoxur and other insecticides was assessed after being subjected to short- and long-term exposure to sublethal doses of propoxur. In addition, the resistance mechanism and the transgenerational effects on *Cx. quinquefasciatus*, especially at the multigenerational level, were analyzed. This study provides valuable information for the management of propoxur resistance and lays a foundation for the development of effective integrated pest management programs.

## 2. Materials and Methods

### 2.1. Mosquitoes

A sensitive *Cx. quinquefasciatus* colony (set as F0) was established from a laboratory-reared colony more than 10 years ago without exposure to any insecticide. The larvae were reared in enamel pans filled with dechlorinated tap water and fed with a mixture of yeast powder and wheat mill. The pupae were removed from the pans every day and placed in cages for emergence. The male adults were fed on 10% sucrose solution, while the females were fed with blood from mice. All larvae and adults were held at 27 ± 1 °C and a photoperiod of 12:12 h (light–dark).

### 2.2. Insecticides

The insecticide propoxur (97%) was supplied by Jiangsu Changlong Agrochemical Co., Ltd Jiangsu, China. Etofenprox (95%), nitenpyram (96%), abamectin (97%), chlorpyrifos (98%), and deltamethrin (95%) were supplied by Hubei Kangbaotai Fine-Chemicals Co., Ltd. Wuhan, China. The spinetoram (60 g/L SC) was supplied by the Dow AgroSciences Company. Propoxur, etofenprox, abamectin, chlorpyrifos, and deltamethrin were dissolved in acetone, and nitenpyram and spinetoram were dissolved in water. Active ingredient (0.1 g) was dissolved in 2 mL acetone, and 50 g/L mother liquor was obtained. Then, it was diluted to 0.1 mg/L, 1 mg/L, 10 mg/L, and 100 mg/L with water. The same volume of solvent without insecticide was added to each control group during the bioassay.

### 2.3. Bioassay

Bioassays were carried out according to the standard method recommended by the World Health Organization [6]. Twenty third-instar larvae were transferred into 100 mL of distilled water in a 200 mL plastic cup. Three replicates for each dose and nine doses in total for each insecticide were set, and the controls were treated with dechlorinated tap water. All treatments were maintained at a temperature of 27 ± 1 °C and 40–50% relative humidity with a 12 h light/12 h dark photoperiod. Mortality was assessed after exposure to insecticides for 24 h.

### 2.4. Resistance Selection and Sublethal Effects of Short-Term Exposure to Propoxur

The LC_25_ (lethal concentration of 25%) and LC_50_ (lethal concentration of 50%) concentrations of propoxur were first determined using the 25% and 50% mortality rates, respectively, of the susceptible strain F0 (not treated by propoxur) assessed after exposure for 24 h. For resistance selection, about 300–2000 third-instar larvae were treated with the LC_25_ and LC_50_ concentrations of propoxur every day to pupa stage, and the mortality rate of the larvae was maintained in the range 20–90% in each generation from F1 to F16. The colonies of F1–F16 were named according to continuous selection by propoxur from generations 1 to 16. In addition, the following data were also observed and recorded daily: the number of eggs and hatching larvae, the number of surviving larvae and pupa, the number of female adults and male adults, and the developmental time of eggs, larvae, pupae, female adults, and male adults.

### 2.5. Assessment of the Transgenerational Effects of Propoxur Exposure

The F1, F5, F10, and F15 colonies of the propoxur-sel strains were developed from the eggs of F0, F4, F9, and F14 treated with LC_25_ and LC_50_ of propoxur and collected on the same day, and they were transferred to an enamel basin with 3000 mL dechlorinated tap water. The first-instar larvae were collected randomly from the newly hatched colonies, individually transferred to an enamel basin with 1000 mL dechlorinated tap water, and reared until to the fourth and final instar. The pupae were collected in a 200 mL beaker with 150 mL distilled water. The adult males and females were put into a cage and were maintained with 10% sucrose solution. After blood feeding on mice for 72 h, the female adults were transferred into another cage and maintained with 10% sucrose solution. Volumes of 150 mL of distilled water in 200 mL plastic cups were put in the cage for their oviposition. The following data were observed and recorded daily: the number of eggs and hatching larvae, the number of surviving larvae and pupae, the number of female adults and male adults, and the developmental time of eggs, larvae, pupae, female adults, and male adults.

### 2.6. Detection of Mutation in AChE Encoding Gene Ace-1

Total RNA was isolated from the fourth-instar larvae of *Cx. quinquefasciatus* using a TRIzol^TM^ Reagent (Thermo Fisher Scientific Life Technologies Corporation, Carlsbad, CA, USA). cDNA synthesis was performed using the GoScriptTM Reverse Transcription System (Promega, Madison, WI, USA). The ace-1 genes of the surviving larvae after exposure to propoxur were amplified with the primers described in [25]. The purified PCR fragments were used for sequencing and comparison with those of the wild population.

### 2.7. Enzyme Activity Assays

To determine the activities of esterase, glutathione *S*-transferase (GST), and cytochrome P450 monooxygenase (P450) of *Cx. quinquefasciatus*, 60 larvae from each generation were homogenized on ice in 1 mL of 0.1 M sodium phosphate buffer (pH 7.0 containing 1 mM EDTA, 1 mM dithiothreitol (DTT), 1 mM phenylthiourea, 1 mM PMSF (phenylmethanesulfonyl fluoride), and 20% glycerol) [26]. The homogenate was then centrifuged at 15,000× *g* for 20 min at 4 °C. The supernatants were harvested as the mosquito crude extractions and stored at −80 °C until use. The protein concentrations were determined using the Bio-Rad Protein Assay Kit, Bio-Rad Laboratories, Hercules, CA, USA.

The esterase activity was determined as previously described with minor modifications. In brief, 200 μL of the assayed mixture which contained 2 μL of α-naphthyl acetate substrate (0.2 mM) and 10 μL of diluted mosquito crude extraction in sodium phosphate buffer (0.2 M, pH 7.0) was pipetted into a 96-well plate and incubated at 37 °C for 15 min. The reaction was stopped by the addition of the colorimetric reagent FAST Blue B, and the absorbance was measured at OD 450 nm.

The glutathione *S*-transferase (GST) activity was assessed using 1-chloro-2, 4-dinitrobenzene (CDNB) as the substrate as previously described. The 200 μL reaction mixture consisted of 6 μL of 30 mM CDNB substrate solution, 6 μL of 30 mM GSH, and 10 μL of the diluted mosquito crude extraction in sodium phosphate buffer (0.1 M, pH 7.0). The absorbance was measured using an ultraviolet spectrophotometer at 340 nm for 5 min with a read interval of 30 s.

The cytochrome P450 monooxygenase (P450) activity was determined using p-nitroanisole (p-NA) as the substrate as previously described [27]. Volumes of 100 μL of 2 mM p-NA, 10 μL of 9.6 mM NADPH, and 90 μL of the diluted mosquito crude extraction in sodium phosphate buffer (0.1 M, pH 7.0) were mixed and then pipetted into a 96-well plate. After incubation at 34 °C for 30 min with shaking, the absorbance was recorded using a microplate reader (Bio-Rad) at 405 nm.

### 2.8. Quantitative Real-Time PCR (qRT-PCR)

Total RNA was isolated from the fourth-instar larvae of *Cx. quinquefasciatus* using a TRIzol^TM^ Reagent (Thermo Fisher Scientific Life Technologies Corporation, Carlsbad, CA, USA). cDNA synthesis was performed using the GoScriptTM Reverse Transcription System (Promega, Madison, USA). The induction of P450 gene expression was performed with the SYBR Green Master Mix Kit on a MyiQ2 real-time PCR system (Bio-Rad, California, USA). The qRT-PCR was in a 25 μL final reaction volume containing 1 × SYBR Green Master Mix, 1 μL of cDNA, and a P450 gene-specific primer pair, designed according to each of the P450 gene sequences as described previously [28,29,30] at a final concentration of 3–5 μM. RT-qPCR was performed with the following cycling regime: initial incubation at 95 °C for 5 min, followed by 40 cycles of 95 °C for 5 s and 55 °C for 10 s. The 18 s ribosome RNA gene was used to normalize the expression of target genes. Relative expression levels for the P450 genes were calculated by the 2^−ΔΔCT^ method [31], in which ΔΔCT = (CT, Target − CT, 18S rRNA)Treated − (CT, Target − CT, 18S rRNA)Control.

### 2.9. Statistical Analysis

The LC_50_ values with 95% confidence interval, slopes with standard error (SE), and chi-square value (χ^2^) with degree of freedom (df) were calculated using a regression model based on a probit transformation of mortalities and a logarithmic transformation of concentrations tested, i.e., a log-probit model by Polo Plus software. The means and standard errors (SE) of the mosquito population parameters of the developmental traits were analyzed by using TWOSEX-MSChart software as previously described [32,33,34,35,36]. The egg hatchability is the percent of eggs that hatched (the number of hatching larvae/the number of eggs). The fecundity (the mean number of eggs laid per adult female of *Cx. quinquefasciatus*), longevity, sex ratio (F (female)/M (male)), population doubling time (DT), and the population parameters (r, the intrinsic rate of increase; λ, the finite rate of increase; R_0_, the net reproductive rate; T, the mean generation time) were calculated accordingly.
(1)$$T=\frac{\ln R_{0}}{r}\;$$
(2)$$R_{0}=\overset{\infty }{\underset{x=0}{\sum }}l_{x}m_{x}$$
(3)$$\mathrm{DT}=\frac{\ln2}{r}$$
(4)$$\lambda =e^{r}$$
(5)$$\overset{\infty }{\underset{x=0}{\sum }}*e^{-r\left(x+1\right)}l_{x}m_{x}=1$$

Here, m_x_ is the average egg number laid per surviving *Cx. quinquefasciatus* during the age interval x; l_x_ is the probability that a newborn will survive from age 0 to x. The mean generation time (T) is the length of time of an insect population increasing to R_0_-fold of its initial size (i.e., e^rT^ = R_0_ or λ^T^ = R_0_) at the stable age structure distribution. The doubling time (DT) is the length of time that an insect population needs to double its initial size (i.e., e^rD^ = 2 or λ^D^ = 2) after the population reaches the stable age structure distribution. The parameters r and λ reveal the effect of insect reproductive ages on the population growth rate at the stable age structure distribution. The parameter R_0_ represents the total offspring number of an average individual.

TWOSEX-MSChart for the age-stage, two-sex life table analysis was used to compare the population parameters (r, λ, R_0_, and T), fecundity, longevity, proportions of male and female (F/M), and population doubling time (DT) differences among treatments by the Tukey Kramer procedure. The data of enzyme activity and expression of P450 were analyzed by ANOVA followed by Duncan’s test. *p* < 0.05 was thought to be statistically significant. The statistical analyses were performed using IBM SPSS Statistics software (v19). However, significant overexpression was determined using a cut-off value of a ≥2-fold change in expression [29].

## 3. Results

### 3.1. Resistance Development After Selection with Sublethal Exposure to Propoxur

The propoxur-sel strains of *Cx. quinquefasciatus* were derived in the laboratory from a lab-feeding strain following 16 generations of propoxur exposure (Figure 1). F16 developed 8.8-fold resistance to propoxur in LC_25_ selection with an LC_50_ value of 2.55 mg L^−1^ and 6.3-fold resistance in LC_50_ selection with an LC_50_ of 1.82 mg L^−1^ compared with the initial LC_50_ of 0.29 mg L^−1^ in the susceptible strain (F0), respectively (Figure 1). However, compared with the susceptible strain, the propoxur-sels showed no significant cross-resistance to the other six tested insecticides (etofenprox, deltamethrin, nitenpyram, abamectin, chlorpyrifos, and spinetoram; Figure 2).

### 3.2. Effects on the Developmental Traits of Adults, Pupae, and Eggs of Parental Cx. Quinquefasciatus after Sublethal and Lethal Exposure to Propoxur

Both the LC_25_ and LC_50_ propoxur treatments obviously affected the mortality rates: the former resulted in mortality rates of 15.6–57.3% in F1–F16 compared with 8.2% in F0, and the latter as high as ca. 60.9–97.9% in F1–F16 compared with 5.0% in F0. In addition, the sex ratios in the adult survivors of propoxur-exposed *Cx. quinquefasciatus* colonies were significantly affected, displaying female/male ratios (F/M) of 1:3–1:6 and 1:4–1:11 in the LC_25_ and LC_50_ treatments, respectively, compared to 1:1 in F0 without the treatment of propoxur (Table 1). Moreover, ca. 6.3–16.2% and 5.3–15.5% reductions in egg hatching of the adult survivors were observed in the LC_25_ and LC_50_ propoxur treatments, respectively. However, the emergence ratio and pupa survival rate were not affected (Table 1).

### 3.3. Transgenerational Impact of Propoxur on Demographic Parameters and Developmental Traits

The differences in demographic parameters and developmental traits between progeny of the untreated susceptible strain (F0) and propoxur-sel strains (F1, F5, F10, and F15) with the LC_25_ and LC_50_ propoxur treatments are shown in Table 2 and Table 3. The longevities and fecundities of propoxur-sel strains decreased nearly 14.3–26.2% and 4.3–12.2%, respectively, compared with that of F0. The sex ratio (F/M) decreased from 1:1 in F0 to 1:2 in F1 and F5 and 1:3 in F10 and F15 in propoxur-sel strains. Correspondingly, the population doubling time (DT) in propoxur-sel strains increased by 29.2–82.1% compared to that in F0. In addition, the intrinsic rates of increase (r) reduced from 0.16 in F0 to 0.10–0.13 in propoxur-sel strains, and the net reproduction rates (R_0_) were also significantly decreased to only 33.4–58.9% that of F0. The finite rate of increase (λ) was decreased by 4.2–7.6% compared with that of F0 as well. Meanwhile, the mean generation time (T) was significantly prolonged in propoxur-sel strains compared to that in F0.

### 3.4. Analysis of the Possible Resistance Mechanism

#### 3.4.1. Metabolic Enzyme Activity

The activities of both esterase and cytochrome P450 monooxygenase (P450) in the selected generations displayed a slight increasing tendency compared with F0 (Figure 3). The former activity level varied from 2.20 ± 0.50 to 4.55 ± 0.43 µmol/min/mg protein in the strains selected by LC_25_ of propoxur and from 2.19 ± 0.17 to 3.54 ± 0.39 µmol/min/mg protein in the populations exposed to LC_50_ of propoxur (Figure 3). The latter activity level ranged from 0.52 ± 0.05 to 0.98 ± 0.14 nmol/min/mg protein in the strains selected by LC_25_ of propoxur and from 0.56 ± 0.02 to 0.82 ± 0.11 nmol/min/mg protein among the populations processed using LC_50_ of propoxur (Figure 3). However, the variation trend of the activities of glutathione *S*-transferase (GST) was irregular among the propoxur-sel populations of *Cx. quinquefasciatus* (Figure 3).

#### 3.4.2. Transcription Levels of P450 Genes in F0 Induced by Propoxur

Significantly higher over-transcription levels of 18 and 16 of the 32 tested P450 genes were found to be induced by the LC_25_ and LC_50_ concentrations compared with the control, respectively, by significance analysis (Figure 4); the 14 genes *cyp9j45*, *cyp4h40*, *cyp6ag12*, *cyp9al1*, *cyp6aa8*, *cyp4d42v1*, *cyp6bz2*, *cyp6aa7*, *cyp6z12*, *cyp9j33*, *cyp4c52v1*, *cyp6aa9*, *cyp4c38*, and *cyp9j43* displayed significant increases under both treatments (Figure 4).

#### 3.4.3. Transcription Levels of P450 Genes in Propoxur-Sel Strains of F1, F5, F10, and F15

In F1 of the strains selected by LC_25_ and LC_50_ levels of propoxur, four genes (*cyp325g4*, *cyp6p14*, *cyp4d42v1*, *cyp4c52v1*) and five genes (*cyp12f13*, *cyp325g4*, *cyp9j45*, *cyp9j43*, *cyp6aa9*), respectively, were significantly over-expressed compared with in the control (Figure 5). Further, the transcription levels of the four genes *cyp325g4*, *cyp4d42v1*, *cyp4c52v1*, and *cyp6aa9* and the seven genes *cyp9j35*, *cyp325g4*, *cyp9j45*, *cyp6p14*, *cyp4d42v1*, *cyp4c52v1*, and *cyp6aa9* in F5 of the strains selected by LC_25_ and LC_50_, respectively, were obviously higher (Figure 5). Moreover, the eight genes *cyp9j35*, *cyp12f13*, *cyp325g4*, *cyp6p14*, *cyp4d42v1*, *cyp4c52v1*, *cyp9j43*, and *cyp6aa9* and the four genes *cyp9j45*, *cyp4d42v1*, *cyp4c52v1*, and *cyp6aa9* in F10 of the strains selected by LC_25_ and LC_50_, respectively, were found to show significant over-transcription (Figure 5). Lastly, in F15 of the strains selected by LC_25_ and LC_50_ levels of propoxur, the expression levels of the eight genes *cyp9j40*, *cyp325g4*, *cyp9j45*, *cyp6ag12*, *cyp9al1*, *cyp4d42v1*, *cyp4c52v1*, and *cyp6aa9* and the seven genes *cyp325g4*, *cyp9j45*, *cyp9j42*, *cyp6p14*, *cyp4d42v1*, *cyp4c52v1*, and *cyp6aa9*, respectively, were higher (Figure 5).

The transcription levels of *cyp4d42v1*, *cyp4c52v1*, and *cyp6aa9* were significantly increasingly transcribed in all three tested *Cx. quinquefasciatus* generations (F5, F10, and F15) selected by either LC_25_ or LC_50_ treatments (Figure 5).

## 4. Discussion

The overreliance on either chemical or biological insecticides for mosquito control has shown the tendency to resistance and physiological alterations developing in mosquito populations [7,8,9]. Therefore, understanding the sublethal and lethal effects of insecticides could be crucial for decision-making in resistance and integrated pest management programs [37]. This study assessed the effects of propoxur, a widely used broad-spectrum carbamate insecticide, on *Cx. quinquefasciatus* at the multigenerational level, including resistance development, metabolic mechanisms, and sublethal effects. The data provide a platform for understanding the potential relationship between the toxicity of a given product in laboratory assays and the exposure risk associated with resistance development under field conditions.

We observed that propoxur exposure at concentrations of LC_25_ and LC_50_ for 16 generations led to low-level resistance (<10-fold) in *Cx. quinquefasciatus*, which corresponds to previous studies in which it was found that the resistance ratios of *Cx. pipiens pallens* could rise up to 2.5- and 7.9-fold after 8 and 16 generations of propoxur selection, respectively [38,39]. Interestingly, the lower concentration (LC_25_) could lead to higher resistance from F10 to F16 when compared with the higher one (LC_50_) (Figure 1). One possible reason for this could be that a low dose could induce multifactorial resistance development at a relatively slow speed, while a high dose could promote evolution of the major resistant gene(s) rapidly. Previous studies showed that the propoxur-resistant strain of *Cx. pipiens pallens* was cross-resistant to DDVP (dichlorvos) and cypermethrin [40,41] but not resistant to DDT, deltamethrin, and acetofenate [42,43,44,45]. Nevertheless, the propoxur-sels displayed cross-resistance to neither deltamethrin nor others like etofenprox, nitenpyram, abamectin, chlorpyrifos, and spinetoram in this study (Figure 2). This indicated that these insecticides could be used in rotation with propoxur in the management of insecticide resistance.

Resistance development as a result of AChE structural modification has been broadly documented in many insects, including several mosquito species. For instance, Zhao et al. (2014) in their study associated the A328S, V185M, and G247S mutations with propoxur resistance in *Cx. quinquefasciatus* [25]. In our current study, no mutation was detected in ace-1 of the propoxur-sel strains (data not shown), which indicated that the propoxur-sels might have developed some other defense mechanism(s). Currently, many studies have shown cytochrome P450 monooxygenase to play an important role in the detoxifying strategies of sanitary pests against propoxur [46,47]. For example, it was reported that propoxur resistance in German cockroach populations was associated with increased activity of cytochrome P450 monooxygenase [46]. In another study, both P450 monooxygenase and esterase were involved in propoxur resistance in field populations of German cockroaches in Singapore [47]. Indeed, enhanced activities of cytochrome P450 monooxygenase and esterase were observed in propoxur-sel strains in this study, indicating that a detoxification mechanism might be responsible for resistance development in *Cx. quinquefasciatus* (Figure 3). In a previous study, the overexpression of the P450 gene *cyp9m10* was suggested to be responsible for permethrin resistance in two isogenic strains of *Cx. quinquefasciatus* (i.e., ISOP450 and ISOJPAL), and eight polymorphic sites of the resistant alleles were found to be different from the susceptible ones [48]. Further, some other P450 genes were also found to be induced (overexpressed) by permethrin in *Cx. quinquefasciatus*, with different sets involved in different resistant strains (e.g., *cyp325k3v1*, *cyp4d42v2*, *cyp9j45*, (*cyp*) CPIJ000926, *cyp325g4*, *cyp4c38*, and *cyp4h40* in the HAmCqG8 strain, and *cyp9m10*, *cyp6z12*, *cyp9j33*, *cyp9j43*, *cyp9j34*, *cyp306a1*, *cyp6z15*, *cyp9j45*, *cyp9al1*, *cyp4c52v1*, and *cyp9j39* in the MAmCqG6 strain) [29]. In field populations of *Cx. pipiens pallens* Coquillett and *Cx. quinquefasciatus*, five up-regulated genes of P450 (*cyp345p1*, *cyp358b1*, *cyp4fd2*, *cyp4cd2*, and *cyp6jn1*) were found to be associated with propoxur resistance in *Cx. pallens* and *Cx. quinquefasciatus* [49]. Since previous studies indicated that most of the P450 genes associated with insecticide resistance belong to the cyp4 and cyp6 clades [29,49,50], we selected 32 P450 genes belonging to these clades for analysis. The data showed that induction by different doses of propoxur and constitutive expression in different generations of propoxur-sel strains resulted in inconsistent identification of the P450 genes probably related with resistance, although the three genes *cyp4d42v1*, *cyp4c52v1*, and *cyp6aa9* displayed a coincidence to some degree in different treatments (Figure 4 and Figure 5). This indicated that the pathway of metabolic detoxification involving cytochrome P450 monooxygenase in mosquito is rather complicated.

Sublethal and lethal effects of insecticides can not only induce direct mortality and promote development of resistance but may also have effects on the development, survival, and reproduction of the target insect [8,15]. This phenomenon has been reported in mosquito and other pests. A study on *Anopheles stephensi* Liston observed that sublethal exposure to propoxur resulted in decrements in fecundity, egg hatchability, and F/M sex ratio but also in prolonged larval duration and adult longevity [14]. In another study, *A. aegypti* and *A. albopictus* exposed to propoxur were also found to live longer and with a heightened fertility level compared with the control [51]. Similarly, this study also showed that the fecundity, longevity, sex ratio (F/M), and hatchability of the adult survivors of propoxur-exposed *Cx. quinquefasciatus* were decreased compared with those of the untreated counterpart (F0). The sublethal effects on physiology and the decrease in fecundity induced by propoxur exposure have also been reported in other insects, including *Musca domestica* and German cockroach (*Blattela germanica*). In *Musca domestica*, a decrease in fecundity on propoxur exposure was discovered [52]. The oothecae could fall off before maturity in *Blattela germanica*, and delayed hatching was observed in the ones that retained the oothecae [53]. When the oothecae were treated with propoxur, the hatching and nymph survival rates were also reduced [53]. Similar physiological defects were also observed in *Cx. quinquefasciatus* with malathion and in *An. quadrimaculatus* with aldrin, chlordane, DDT, BHC (benzene hexachloride), and rotenone [37,54]. These sublethal effects occurred probably because coping with the toxicity of insecticides could be dire and could require energy and resource allocation for adaptation and survival. The demographic parameters (r, Ro, T, and λ) and sex ratio of the *Cx. quinquefasciatus* offspring also showed a significant difference compared to F0. In a previous report, significant sex ratio distortion was only found to occur in the parental generation [37]. However, this study showed a significant sex ratio distortion not only in the parental generation but also in the progeny of *Cx. quinquefasciatus*, which might have occurred through vertical transfer of propoxur from the mother to the offspring.

## 5. Conclusions

In general, when exposed to sublethal and lethal doses of propoxur, *Cx. quinquefasciatus* could develop resistance via physiological adaption. The application of this insecticide could lead to a decline of the population not only by killing susceptible ones but also by reducing the reproductive potential among the resistant strains and, finally, impacting the population density of mosquito communities. Thus, this study provides a reference for the rational application of propoxur for mosquito control in the field.

## Figures and Tables

**Figure 1 insects-10-00288-f001:**
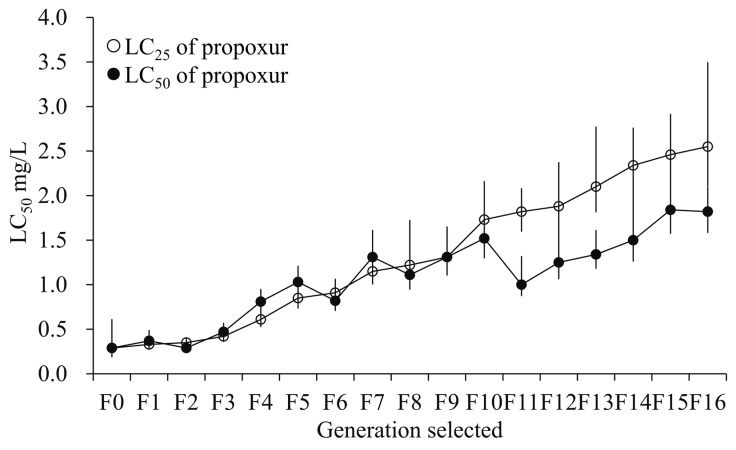
Resistance development of *Culex pipiens quinquefasciatus* after selection with LC_25_ (lethal concentration of 25%) (open circles) and LC_50_ (lethal concentration of 50%) (shaded circles) levels of propoxur.

**Figure 2 insects-10-00288-f002:**
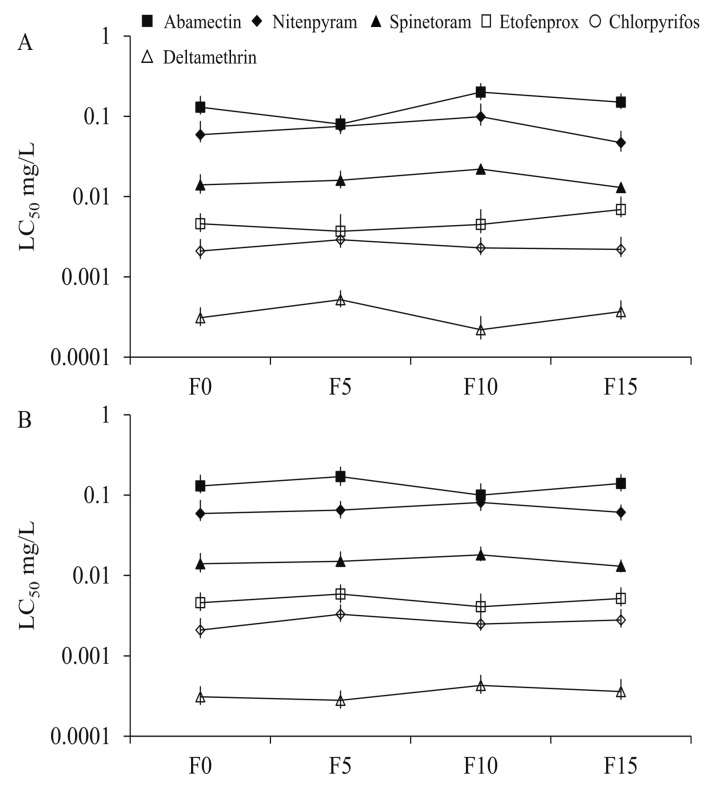
Cross-resistance to tested insecticides in *Cx. quinquefasciatus* treated with LC_25_ (**A**) and LC_50_ (**B**) levels of propoxur.

**Figure 3 insects-10-00288-f003:**
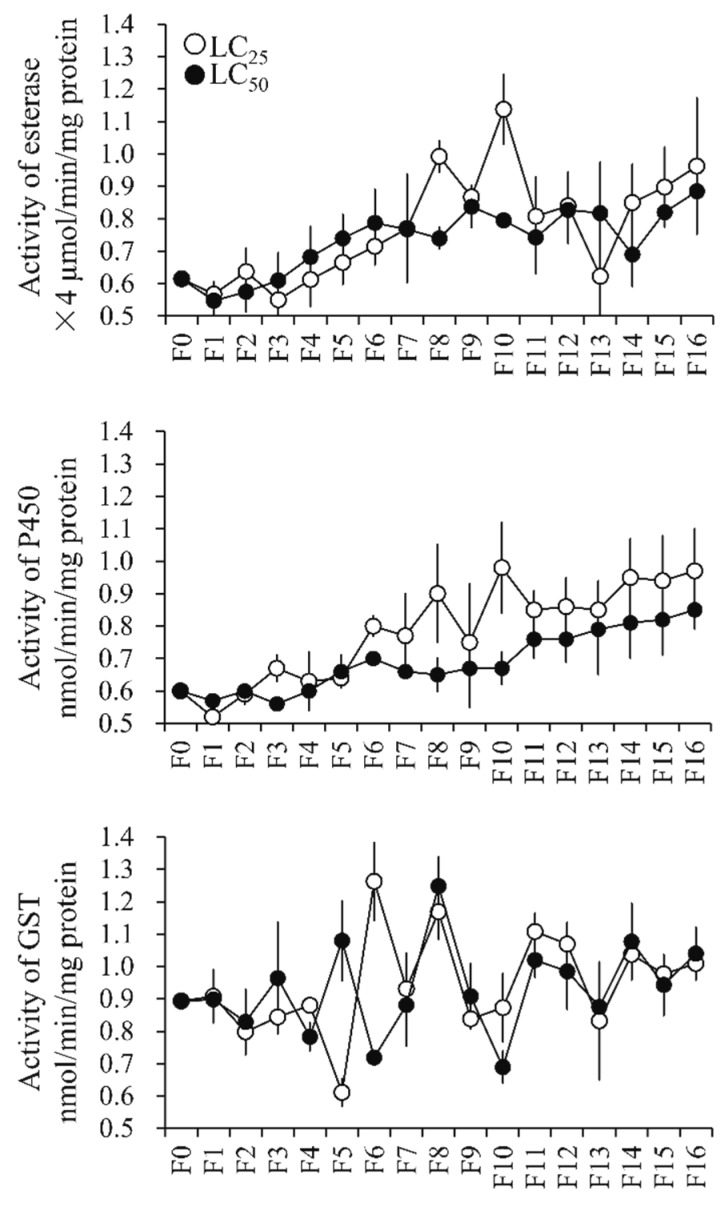
Activity of metabolic detoxifying enzymes in the populations exposed to LC_25_ and LC_50_ levels of propoxur. GST: glutathione *S*-transferase; P450: cytochrome P450 monooxygenase.

**Figure 4 insects-10-00288-f004:**
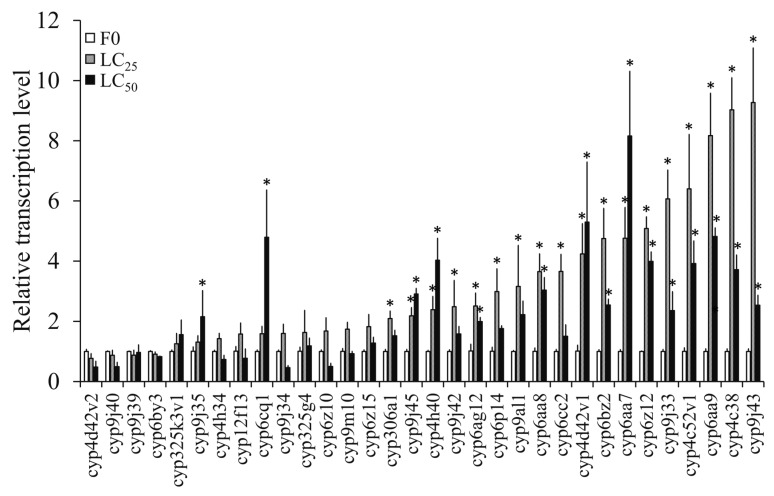
The induction of P450 gene transcription in *Cx. quinquefasciatus* following LC_25_ and LC_50_ exposure to propoxur. The asterisks indicate genes significantly over-expressed compared with in F0 (≥2 folds).

**Figure 5 insects-10-00288-f005:**
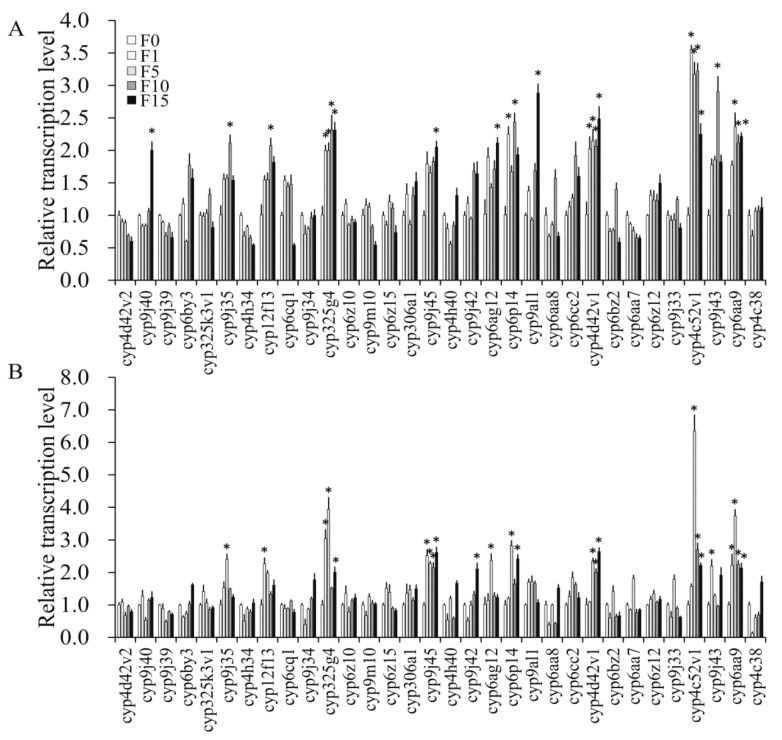
The relative transcription levels of P450 genes in F1, F5, F10, and F15 resistant strains of *Cx. quinquefasciatus* selected by LC_25_ (A) and LC_50_ (B) exposure to propoxur. The asterisks indicate genes which were significantly over-expressed compared with in F0 (≥2 folds).

**Table 1 insects-10-00288-t001:** Effects of prolonged sublethal and lethal exposure to propoxur of *Cx. quinquefasciatus*.

G	*Cx. quinquefasciatus* Treated with LC_25_ of Propoxur				*Cx. quinquefasciatus* Treated with LC_50_ of Propoxur			
	F/M	H%	Sp%	Er%	F/M	H%	Sp%	Er%
F0	1:1 a	92.6 ± 2.6 a	93.8 ± 1.2 a	92.2 ± 1.2 a	1:1 a	93.6 ± 0.1 a	98.7 ± 0.4 a	98.7 ± 0.4 a
F1	1:5 b	91.0 ± 0.8 a	92.6 ± 1.2 a	89.0 ± 1.3 a	1:8 b	87.3 ± 1.2 b	95.7 ± 2.1 a	93.8 ± 3.4 a
F2	1:6 b	87.3 ± 1.8 b	93.6 ± 0.9 a	91.1 ± 1.2 a	1:9 b	85.6 ± 0.4 b	96.3 ± 3.7 a	92.6 ± 7.4 a
F3	1:4 b	85.9 ± 2.4 b	94.6 ± 0.8 a	90.9 ± 0.6 a	1:11 b	83.8 ± 2.8 b	90.8 ± 1.4 b	90.2 ± 1.0 a
F4	1:4 b	81.3 ± 0.2 b	95.0 ± 1.3 a	92.1 ± 1.1 a	1:6 b	82.3 ± 1.0 b	98.3 ± 0.9 a	97.5 ± 1.3 a
F5	1:3 b	83.8 ± 3.1 b	94.3 ± 2.1 a	90.8 ± 1.3 a	1:6 b	84.7 ± 1.1 b	96.9 ± 3.1 a	92.1 ± 4.2 a
F6	1:4 b	84.1 ± 2.3 b	94.1 ± 0.9 a	90.5 ± 1.4 a	1:4 b	82.4 ± 5.3 b	96.4 ± 0.2 a	95.2 ± 1.1 a
F7	1:5 b	84.9 ± 0.8 b	94.8 ± 1.7 a	92.9 ± 2.1 a	1:7 b	81.6 ± 0.8 b	94.6 ± 2.8 a	94.6 ± 2.8 a
F8	1:5 b	80.6 ± 0.6 b	91.2 ± 1.8 a	88.9 ± 2.3 a	1:8 b	86.3 ± 2.4 b	97.8 ± 1.2 a	95.1 ± 1.3 a
F9	1:6 b	81.2 ± 2.0 b	93.0 ± 0.6 a	89.8 ± 0.6 a	1:6 b	80.1 ± 1.8 b	97.2 ± 1.8 a	95.1 ± 3.9 a
F10	1:6 b	77.1 ± 1.7 b	93.1 ± 1.3 a	88.0 ± 2.4 a	1:6 b	82.1 ± 0.2 b	97.4 ± 1.6 a	94.2 ± 0.7 a
F11	1:5 b	79.6 ± 1.6 b	93.9 ± 1.7 a	87.9 ± 1.1 a	1:9 b	84.5 ± 2.8 b	100.0 ± 0.0 a	87.1 ± 11.4 a
F12	1:5 b	81.4 ± 2.3 b	94.8 ± 1.2 a	89.7 ± 2.5 a	1:5 b	84.9 ± 0.6 b	94.0 ± 1.1 a	89.2 ± 2.1 a
F13	1:4 b	81.7 ± 1.4 b	94.3 ± 2.3 a	90.5 ± 2.0 a	1:5 b	79.8 ± 2.8 b	97.8 ± 2.2 a	97.8 ± 2.2 a
F14	1:3 b	82.8 ± 1.0 b	93.0 ± 0.6 a	90.1 ± 0.4 a	1:10 b	82.9 ± 1.3 b	98.2 ± 1.8 a	96.7 ± 1.7 a
F15	1:3 b	82.1 ± 1.2 b	94.1 ± 1.5 a	90.8 ± 1.7 a	1:4 b	77.4 ± 3.2 b	94.3 ± 3.2 a	87.7 ± 2.8 a
F16	1:5 b	84.8 ± 1.5 b	94.3 ± 0.3 a	90.8 ± 0.8 a	1:4 b	82.2 ± 2.2 b	94.8 ± 2.6 a	89.3 ± 1.8 a

G: generation, F/M: proportion of males and females, H: egg hatchability, Sp: survival rate of pupae (the number of surviving pupae on the first day/number of pupation), Er: emergence rate. The values are given as mean ± SE. The effects of prolonged sublethal and lethal exposure were analyzed by ANOVA followed by Duncan’s test using IBM SPSS Statistics 19. a: No significant difference compared with F0 at the *p* = 0.05 level. b: Significant difference compared with F0 at the *p* = 0.05 level.

**Table 2 insects-10-00288-t002:** Transgenerational effects of exposure to propoxur to doubling time, fecundity, longevity, and sex ratio of *Cx. quinquefasciatus*.

G	*Cx. quinquefasciatus* Treated with LC_25_ of Propoxur				*Cx. quinquefasciatus* Treated with LC_50_ of Propoxur			
	DT (d)	Fecundity (eggs/♀)	Longevity (d)	F/M	DT (d)	Fecundity (eggs/♀)	Longevity (d)	F/M
F0	4.25 ± 0.11 a	70.22 ± 2.84E-14 a	39.49 ± 1.31 a	1:1 a	4.25 ± 0.11 a	70.22 ± 2.84E-14 a	39.49 ± 1.31 a	1:1 a
F1	5.49 ± 0.25 b	70.33 ± 1.42E-14 a	31.48 ± 1.32 b	1:2 b	6.45 ± 0.35 b	63.44 ± 4.26E-14 b	31.18 ± 1.36 b	1:2 b
F5	6.08 ± 0.32 b	66.77 ± 2.84E-14 b	33.83 ± 1.42 b	1:2 b	6.78 ± 0.52 b	64.80 ± 3.89E-14 b	33.11 ± 1.50 b	1:3 b
F10	6.79 ± 0.43 b	64.35 ± 1.42E-14 b	31.55 ± 1.31 b	1:3 b	7.04 ± 0.56 b	61.66 ± 1.42E-14 b	31.79 ± 1.43 b	1:3 b
F15	6.46 ± 0.41 b	67.19 ± 2.84E-14 b	29.99 ± 1.23 b	1:3 b	7.74 ± 0.71 b	66.96 ± 1.42E-14 b	29.13 ± 1.54 b	1:3 b

DT: Population doubling time. The values are given as mean ± SE. a: No significant difference compared with F0 at the *p* = 0.05 level. b: Significant difference compared with F0 at the *p* = 0.05 level.

**Table 3 insects-10-00288-t003:** Transgenerational effects of propoxur exposure on demographic parameters of *Cx. quinquefasciatus.*

G	*Cx. quinquefasciatus* Treated with LC_25_ of Propoxur				*Cx. quinquefasciatus* Treated with LC_50_ of Propoxur			
	r (d^−1^)	R_0_	T (d)	λ (d^−1^)	r (d^−1^)	R_0_	T (d)	λ (d^−1^)
F0	0.16 ± 0.0043 a	29.52 ± 2.61 a	20.76 ± 0.00072 a	1.18 ± 0.0056 a	0.16 ± 0.043 a	29.52 ± 2.61 a	20.76 ± 0.00072 a	1.18 ± 0.0051 a
F1	0.13 ± 0.0055 b	17.40 ± 2.15 b	22.62 ± 0.00095 b	1.13 ± 0.0063 b	0.10 ± 0.0053 b	14.59 ± 1.95 b	25.70 ± 0.11 b	1.11 ± 0.0059 b
F5	0.11 ± 0.0058 b	14.19 ± 1.90 b	23.28 ± 0.00059 b	1.12 ± 0.0065 b	0.10 ± 0.0073 b	10.86 ± 1.81 b	23.34 ± 0.00081 b	1.11 ± 0.0081 b
F10	0.10 ± 0.0061 b	12.27 ± 1.81 b	24.57 ± 0.00013 b	1.11 ± 0.0068 b	0.10 ± 0.0073 b	9.99 ± 1.67 b	23.40 ± 0.0019 b	1.10 ± 0.0080 b
F15	0.11 ± 0.0065 b	12.31 ± 1.83 b	23.40 ± 0.0011 b	1.11 ± 0.0072 b	0.10 ± 0.0074 b	9.85 ± 1.81 b	25.55 ± 0.028 b	1.09 ± 0.0081 b

r: Intrinsic rate of increase, R_0_: Net reproductive rate (offspring per individual), T: Mean generation time, λ: Finite rate of increase. The values are given as mean ± SE. a: No significant difference compared with F0 at the *p* = 0.05 level. b: Significant difference compared with F0 at the *p* = 0.05 level.

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
