# Peer review of "Effects of Propoxur Exposure on Insecticidal Susceptibility and Developmental Traits in Culex pipiens quinquefasciatus"

_insects, 2019, doi:10.3390/insects10090288_

Round 1

Reviewer 1 Report

The manuscript by Zhang and colleagues reports the effect a carbamate compound (propoxur) and its effects on different generations of a mosquito species (Culex pipiens quinquefasciatus), and looked at gene expression when exposed to sublethal concentrations of the pesticide. A whole series of genes related to P450 expression have been quantified under exposure to sublethal concentrations. The results reported an increase in the LC50 for the next generations, an increase in mortality rates, and a bias toward females in the male/female sex ratio. Other parameters have been observed and reported. For the gene expression, about half of the P450 genes were upregulated with the exposure to sublethal dose of propoxur. The discussion tries to link these results with those obtain in other insect species (esp. cockroach), and points out the differences with expression of P450 genes in other mosquito species. They conclude with by promoting propoxur as a good compound to kill susceptible mosquitoes and to reduce reproductive potential of the resistant strains.

The manuscript is rather well written scientifically sound enough, but it feels a few things are out of place, and some results are unclear, which makes the discussion a bit lengthy and confusing. These different points are listed below, and makes me recommend this paper for major revision.

Main Comments

My first main comment is about the mortality rates of the F0 strain at LC25 and LC50 concentrations. I might understand that wrong but I would expect the LC25 concentration to kill about 25% of the susceptible strain F0, and the LC50 concentration to kill about 50%. But, in the text and in the table, it is reported to kill 8.2% and 5.0%. I do not understand, and it is the only instance where it is happening. Every subsequent generation has a higher mortality rate with the LC25 and the LC50. I looked in the methodology but there is no mention of any range of concentration you tested (for any pesticide), or how you determined the mortality. Can you please clarify this point in the methods and in the results?

The second main comment is about the gene expression levels and the Figures 4 and 5. Beside the LC25 data in Figure 4, these data are not easy to read, and you do not help the reader to walk them through. I don’t see what Figure 5A brings, for instance. In the Results, section 3.4.2, you wrote “significant increasing under both treatments”. Was there any stat analysis done for this data? If so, please make it appear in the Figure. If not, please do not use the word “significant”. In a similar way, the next section, you wrote “the transcription levels […] were obviously higher in both strains”. I do not see anything obvious, unless you explain it through the figures. Can you please modify accordingly?

Other comments

Introduction (line 32-35): long sentence, please split it in two sentences.

Introduction (line 44-45): the sentence “Thus, mosquitoes are likely to be sublethal dose exposure during insecticide application in the control management” is phrased in an odd way. Do you mean to say that the mosquitoes are likely to exposed to sublethal doses during the control application of pesticides? Please rephrase the sentence.

Introduction (line 53): “and so on” is unclear. Please either define it, or remove that part.

Introduction (line 66-67): if propoxur is more and more detected in the environment, is it a good thing to continue using it?

Methods (2.3): please specify in which solvent you diluted your pesticide compounds listed in section 2.2, whether it’s water, ethanol, or another. Consequently, did you put the same amount of solvent in the control group?

Methods (2.4 and 2.5): please explain what is hatchability and developmental time, and define how you control fecundity.

Results (3.1 and Figure 2): In the Figure 1, you use regular units for the Y-axis, and in Figure 2, you use the log value. Please use one or the other, but keep both figures consistent.

Results (3.4.1 and Figure 3) you use “most significant increasing” twice in this section, but there are no stats related to that. Please use a statistical test, or rephrase accordingly. Also, you mentioned increases for F8 and F10, but you do not talk about the decrease in esterase activity from F10 to F13? Is there any reason/suggestion for that drop? About the figure itself, the Y axes are really confusing. Please find a way to make them clear and consistent between the three graphs.

Results (Table 1) Are the value on the second lines of each generation CI95% (like described in the data analysis section)? If so, please make it appear somewhere.

Results (Table 2 and 3) similar to the previous comment, what does the second line of each generation represent? Also, is the fecundity in %? what are the units for the longevity parameters?  Similarly, what is the unit of the “mean generation time”?

Discussion: general comment, please refer to your figures when you can or have to. For instance, Figure 1 after the sentence line 285-287.

Discussion (line 285): Please replace “selecting” by “selection”.

Discussion (line 293-294): the general conclusion of you second paragraph is only based on the fact that the LC50 value of other compounds does not vary much when tested every five generations. Have you looked at the P450 gene expression pattern of these propoxur-selected strains treated with other pesticides? In other words, if the expression level of P450 genes stays similar to the control (18S-RNA) for the propoxur-selected strains exposed to anything else, then your conclusion is valid. If you have not tested it, you can’t say it may be different.

Discussion (line 300-305): you mention the studies done on cockroaches. What are the concrete links between cockroach and Culex in terms of physiology and metabolism that would justify the P450 genes expression?

Discussion (line 326-350): this last paragraph shows how everything you look around P450 gene expression can go in every direction, especially in different mosquito species. It is definitely a strong point to share and discuss, but I am not sure if it only shows that P450 expression is very dependent of the species, and possibly of the environmental factors. This paragraph is slightly undermining your own results. Perhaps you may want to describe the similarities and differences in term of physiology and detoxification processes of close species?

Author Response

Dear reviewer,

Thank you very much for taking the time to review our manuscript and provide kind comment on that. The manuscript has been modified as suggested. The answers to all points, item by item, are also provided.

My first main comment is about the mortality rates of the F0 strain at LC25 and LC50 concentrations. I might understand that wrong but I would expect the LC25 concentration to kill about 25% of the susceptible strain F0, and the LC50 concentration to kill about 50%. But, in the text and in the table, it is reported to kill 8.2% and 5.0%. I do not understand, and it is the only instance where it is happening. Every subsequent generation has a higher mortality rate with the LC25 and the LC50. I looked in the methodology but there is no mention of any range of concentration you tested (for any pesticide), or how you determined the mortality. Can you please clarify this point in the methods and in the results?

Author's response

Yes, the concentrations of LC25 and LC50 of propoxur were first determined by the 25% and 50% mortality rate of the susceptible strain F0 (not treated by propoxur), respectively. However, they are assessed after exposure to 24h, while in the resistance selection, the mosquito larvae were treated with the LC25 and LC50 concentrations of propoxur every day, respectively, until to pupa stage. That is why the final counted mortality rates listed in Table 1 in the old version were lower. The detail has been provided in line 101 and in the line 105-107 in page 3 in the new version. However, the mortalities have been deleted in Table 1 to avoid misunderstandings.

The second main comment is about the gene expression levels and the Figures 4 and 5. Beside the LC25 data in Figure 4, these data are not easy to read, and you do not help the reader to walk them through. I don’t see what Figure 5A brings, for instance. In the Results, section 3.4.2, you wrote “significant increasing under both treatments”. Was there any stat analysis done for this data? If so, please make it appear in the Figure. If not, please do not use the word “significant”. In a similar way, the next section, you wrote “the transcription levels […] were obviously higher in both strains”. I do not see anything obvious, unless you explain it through the figures. Can you please modify accordingly?

Author's response

The data of fig. 4 and fig. 5 have been reanalyzed by ANOVA followed by the Duncan's test. P < 0.05 was thought to be statistically significant. And the data was reanalyzed by using IBM SPSS Statistics software, version 19.0. However, significant overexpression was determined using a cut-off value of a ≥2-fold compared with F0 [29]. The related description has been added in the line 209-213. Furthermore, the genes with significant over-expression have been marked with an asterisk compared with genes in the control (> 2-fold) in Fig4 and Fig5. The related description has ben added in the new version.

Introduction (line 32-35): long sentence, please split it in two sentences.

Author's response

It has been split as suggested in the line 32-35.

Introduction (line 44-45): the sentence “Thus, mosquitoes are likely to be sublethal dose exposure during insecticide application in the control management” is phrased in an odd way. Do you mean to say that the mosquitoes are likely to exposed to sublethal doses during the control application of pesticides? Please rephrase the sentence.

Author's response

The sentence has been rephrased as suggested in the line 44-45.

Introduction (line 53): “and so on” is unclear. Please either define it, or remove that part.

Author's response

Removed as suggested.

Introduction (line 66-67): if propoxur is more and more detected in the environment, is it a good thing to continue using it?

Author's response

The sentence has been rephased as Consequently, it has been frequently detected in agricultural and aquatic environment due to its massive use.” in the line 66-67.

Methods (2.3): please specify in which solvent you diluted your pesticide compounds listed in section 2.2, whether it’s water, ethanol, or another. Consequently, did you put the same amount of solvent in the control group?

Author's response

The description “Propoxur, etofenprox, abamectin, chlorpyrifos and deltamethrin were dissolved in acetone, and nitenpyram and spinetoram were dissolved in water. Active ingredient (0.1 g) was dissolved in 2 ml acetone, and 50 g/L mother liquor was obtained. Then it was diluted into 0.1 mg/L, 1 mg/L, 10 mg/L and 100 mg/L by water. The same volume of solvent without insecticide was added to each control group during the bioassay.” has been added in the new version as suggested in the line 89-94.

Methods (2.4 and 2.5): please explain what is hatchability and developmental time, and define how you control fecundity.

Author's response

The egg hatchability is the percent of eggs that hatched and is calculated as: the egg hatchability = the numbers of hatching larvae / the number of eggs. The fecundity is the mean number of offspring produced per surviving mosquito during the age interval x. The fecundity = the total eggs laid by females / the number of female adults. The development time was observed from egg until to the death of the last adult. And the definition and formula about parameter of life table have been added in the line 176-205. And the sentence “In addition, the following data were observed and recorded: the hatchability, the developmental time, the emergence and duration of adult stage, and the fecundity.” in the line 101-103 in the old version was replaced by the sentence “In addition, The following data were observed and recorded daily: the number of eggs and hatching larva, the number of survival larva and pupa, the number of female and male, the developmental time of egg-adult.” in the line 109-112 in the new version.

Results (3.1 and Figure 2): In the Figure 1, you use regular units for the Y-axis, and in Figure 2, you use the log value. Please use one or the other, but keep both figures consistent.

Author's response

The Y-axis of Fig. 1 and Fig. 2 have been edited as suggested.

Results (3.4.1 and Figure 3) you use “most significant increasing” twice in this section, but there are no stats related to that. Please use a statistical test, or rephrase accordingly. Also, you mentioned increases for F8 and F10, but you do not talk about the decrease in esterase activity from F10 to F13? Is there any reason/suggestion for that drop? About the figure itself, the Y axes are really confusing. Please find a way to make them clear and consistent between the three graphs.

Author's response

The reanalysis has been done, and found that there was not significant difference. The sentences have been rephrased in the line 251-259, and the Y axes of three graphs (Fig. 3) have been edited to keep consistent.

Results (Table 1) Are the value on the second line of each generation CI95% (like described in the data analysis section)? If so, please make it appear somewhere.

Author's response

The value on the second line of each generation were not CI 95%. They were standard error. And the sentence “The values followed ± are mean (SE).” was added in the line 353 in the legend.

Results (Table 2 and 3) similar to the previous comment, what does the second line of each generation represent? Also, is the fecundity in %? what are the units for the longevity parameters?  Similarly, what is the unit of the “mean generation time”?

Author's response

The value on the second line of each generation represent standard error. The sentence “The values followed ± are mean (SE).” was added in the line 362 and 367 in the text. And the sentence “the raw data were analyzed using the age stage and two-sex life table theory to estimate the means and standard errors of the life table parameters by using the bootstrap procedure with bootstrap number n= 200,000 to ensure more precise estimates” was added in the line 176-179 in the text. The units for the longevity and mean generation time were days (d) and it was added in the Table 2. The fecundity was eggs/♀ and it has been added in Table 2. The mean generation time (T) is defined as the length of time that a population needs to increase to some fold of its size at the stable age-stage distribution, and is calculated as T = (ln R0)/r. The units of mean generation time (T), the population doubling time (DT), longevity, intrinsic rate of increase (r), finite rate of increase (λ) have been added in Table 1 and Table 2. And the definition and formula of the population parameters (r, λ, R0, and T), egg hatchability, fecundity, longevity, proportion of population doubling time (DT) and so on have been added in statistical analysis in the line 173-205.

Discussion: general comment, please refer to your figures when you can or have to. For instance, Figure 1 after the sentence line 285-287.

Author's response

Thank you very much for kind comment. Figure 1 has been referred in the sentences in the line 216-220 and in the line 386-388 in new version. Figure 2 has been referred in the sentences in the line 220-223 and in the line 392-394. Figure 3 has been referred in the sentences in the line 251-259 and in the line 406-408. Figure 4 has been referred in the sentences in the line 261-265 and in the line 421-425. Figure 5 has been referred in the sentences in the line 267-282 and in the line 421-425.

Discussion (line 285): Please replace “selecting” by “selection”.

Author's response

Done.

Discussion (line 293-294): the general conclusion of you second paragraph is only based on the fact that the LC50 value of other compounds does not vary much when tested every five generations. Have you looked at the P450 gene expression pattern of these propoxur-selected strains treated with other pesticides? In other words, if the expression level of P450 genes stays similar to the control (18S-RNA) for the propoxur-selected strains exposed to anything else, then your conclusion is valid. If you have not tested it, you can’t say it may be different.

Author's response

The text (line 293-294) in the old version is inappropriate and has been deleted.

Discussion (line 300-305): you mention the studies done on cockroaches. What are the concrete links between cockroach and Culex in terms of physiology and metabolism that would justify the P450 genes expression?

Author's response

There were probably not concrete links between cockroach and Culex in terms of physiology and metabolism that would justify the P450 genes expression. Our discussion was just used to discuss that there might be some link between P450 genes expression in the detoxifying strategies of sanitary pests against propoxur.

Discussion (line 326-350): this last paragraph shows how everything you look around P450 gene expression can go in every direction, especially in different mosquito species. It is definitely a strong point to share and discuss, but I am not sure if it only shows that P450 expression is very dependent of the species, and possibly of the environmental factors. This paragraph is slightly undermining your own results. Perhaps you may want to describe the similarities and differences in term of physiology and detoxification processes of close species?

Author's response

Yes, we just want to describe the similarities and differences in term of physiology and detoxification processes of close species. And some sentences have been revised in the line 428-429.

Reviewer 2 Report

General comments: The manuscript “Effects of propoxur exposure on insecticidal susceptibility and developmental traits in Culex pipiens quinquefasciatus” meets in the present form not the criteria to be published in the journal. The authors evaluated a wide range of effects based on the sub-lethal exposure to propoxur. The main reasons for rejecting the manuscript are inappropriate statistics and language deficits. It would be more conclusive if the wild type Cx. quinquefasciatus would have been tested together with the treated populations to avoid laboratory effects. It is strange that the differences between F0 and F1 are usually great and remain frequently stable to F15. It is proposed to use proof reading and give more details on statistical significance and to submit the manuscript again.

Special comments (here are only a few examples given):

Line 14: Abbreviation of Culex is “Cx.” and “Ae.” for Aedes. Please check the whole manuscript.

Line 40….breeding sites are often aquatic….. breeding sites are always “aquatic”

Line 51…. studies built an association between… not appropriate English

Line 63… warehousing pest control… not appropriate English

Line 81….sieve cages… what is sieve cage?

Line 98…. The mortality was remained among 20-90% in every generation… not appropriate English

Line 305..synergism research…. What is meant with synergism research?

Not conclusive statistical evaluation:

Fig 1: the SD (bars) are more or less always the same.
fig. 2: no standard bars.
fig. 4 and 5: Is it correct that all SD bars are very similar?

Tab. 1: M%: F0 mortality is 5% and F1 (91.9%) to F16 (88,1%), it is strange that within one generation such a quick change occurred and stayed at the same level till F16

Tab. 2: Longevity reduced tremendously from F0 to F1 and stayed then almost constant to F15. Is this correct?

Author Response

Dear reviewer, 

Thank you very much for kind comment on that. The text has been modified as suggested. Specific modification details are as follows.

The manuscript “Effects of propoxur exposure on insecticidal susceptibility and developmental traits in Culex pipiens quinquefasciatus” meets in the present form not the criteria to be published in the journal. The authors evaluated a wide range of effects based on the sub-lethal exposure to propoxur. The main reasons for rejecting the manuscript are inappropriate statistics and language deficits. It would be more conclusive if the wild type Cx. quinquefasciatus would have been tested together with the treated populations to avoid laboratory effects. It is strange that the differences between F0 and F1 are usually great and remain frequently stable to F15. It is proposed to use proof reading and give more details on statistical significance and to submit the manuscript again.

Author's response

For your questions, 1) It is indeed a pity that the current manuscript only focused on the laboratory feeding but not on the field strain of Cx. quinquefasciatus. We had meant to avoid the effect of unknown environmental factors. 2) When selecting the resistant generations, F0 was divided into two populations (F0 and F0’): F0 was set as the control which was not treated by propoxur; F0’ was treated by propoxur from 3rd instar larva to adult. F1 to F16 was also treated by propoxur until to the adult stage. That is why the differences between F0 and F1 were great. 3) The statistical analysis was reperformed in the new version. And the related text has been rewritten in the line 173-213. And Fig.1, Fig.2, Fig. 4 and Fig. 5 have been redrawn. The inappropriate statistics and language deficits have been modified as suggested.

Line 14: Abbreviation of Culex is “Cx.” and “Ae.” for Aedes. Please check the whole manuscript.

Author's response

Abbreviation of Culex has been changed to “Cx.” in the whole manuscript.

Line 40….breeding sites are often aquatic….. breeding sites are always “aquatic”

Author's response

The word “often” has been replaced by the word “always” in the line 39.

Line 51…. studies built an association between… not appropriate English

Author's response

The sentence in the line 50-53 has been modified. It was changed to “And a correlation has been established between the degree of the resistance and enzyme activity expressed in mosquito in previous studies, including esterase, glutathione S-transferase (GST), cytochrome P450 monooxygenase.”

Line 63… warehousing pest control… not appropriate English

Author's response

Modified.

Line 81….sieve cages… what is sieve cage?

Author's response

The “sieve cages” has been modified to “cages” in the line 81.

Line 98…. The mortality was remained among 20-90% in every generation… not appropriate English

Author's response

The sentence has been rephrased as “…the mortality rate was maintained among 20-90% in every generation from F1 to F16.” in the line 107.

Line 305..synergism research…. What is meant with synergism research?

Author's response

The words “synergism research” have been deleted in the line 404-406. The synergism research was conducted for resistance mechanisms of the insecticide resistance strain. The synergists include triphenyl phosphate (TPP), diethyl maleate (DEM) and piperonyl butoxide (PBO). And TPP, DEM and PBO were inhibitor of esterase, glutathione S-transferase (GST), cytochrome P450 monooxygenase, respectively. In synergism research, TPP, DEM, and PBO were applied to larvae or adult through topical application hours before insecticide treatment.

Fig 1: the SD (bars) are more or less always the same

Author's response

We used mean ± SE (standard errors) but not SD. In the old version, there was some mistake in Fig.1. Currently, the standard errors have been modified, and they have been added in the new Fig. 1.

fig. 2: no standard bars.

Author's response

The standard errors (SE) have been added in Fig. 2 in the new version.

fig. 4 and 5: Is it correct that all SD bars are very similar?

Author's response

These standard errors do look similar in Fig. 4 and Fig. 5, especially in Fig. 5. In the calculation process of the relative expression levels for the P450 genes, only the similar threshold cycle for target amplification in repeats of target gene were used to calculate the relative expression level of genes. The result showed that most of the standard errors were from 0.020 to 0.10 or from 0.10 to 0.20 in Fig. 5. Thus, they do appear similar.

Tab. 1: M%: F0 mortality is 5% and F1 (91.9%) to F16 (88,1%), it is strange that within one generation such a quick change occurred and stayed at the same level till F16

Author's response

I am sorry that it was not clear in the description. F0 is the control, it was not treated by propoxur. The larva mortality of F1 to F16 was assessed after exposure until to the pupa stage, thus, the mortality rate with the LC25 and the LC50 have a quick change compared with F0. The detailed description has been added in the line 101 and in the line 105-108. However, the mortalities have been deleted in the Table 1 to avoid misunderstandings.

Tab. 2: Longevity reduced tremendously from F0 to F1 and stayed then almost constant to F15. Is this correct?

Author's response

The same response as above.

Reviewer 3 Report

Congratulation to the authors !

Well done research study and the paper. 

Author Response

Dear reviewer, 

Thank you very much for kind comment on that. The text has been modified as suggested. Specific modification details are as follows.

Author's response

The words “and etc.,” have been deleted.

Author's response

The words “and so on” have been deleted in the line 53.

Author's response

The words “warehousing pest” have been replaced by “store house” in the line 63.

Author's response

The words “of which” have been replaced by the word “while” in the line 82.

Author's response

The word “male” has been inserted between “The” and “adult” in the line 82.

Author's response

This sentence “The mortality was remained among 20-90% in every generation.” has been rewritten to make a new sentence that “While for resistance selection, about 300-2,000 3rd-instar larvae were treated with the LC25 and LC50 concentrations of propoxur very day, respectively, until to pupa stage and the mortality rate was maintained in the range 20-90% in every generation from F1 to F16.” in the line 105-107.

Author's response

The word “among” has been replaced by the words “in the range” in the line 107.

Author's response

The word “they” in the line 115 has been deleted.

Author's response

The words “the 4th-instar stage” have been replaced by the words “the final 4th-instar” in the line 118-119.

Author's response

The sentence in the line 113-115 in the old version has been deleted.

Author's response

The word “population” has been inserted after the words “the wild” in the line 132.

Author's response

The word “()” has been deleted in the line 137.

Author's response

The word “g” has been changed to “× g” in the line 138.

Author's response

The comma has been replaced by the full stop in the line 139. And the words “and the” have been deleted in the line 139.

Author's response

The word “as” has been deleted in the line 148.

Author's response

The sentence has been modified in the line 251-259. And the sentence “in which the most significant increasing was observed in F8 and F10 selected by LC25 of propoxur as well.” has been deleted.

Author's response

The sentence has been modified in the line 251-259. And the sentence “in which the most significant increasing was observed in F8 and F10 selected by LC25 of propoxur as well.” has been deleted.

Author's response

The words “except cyp9j43, cyp9al1, cyp9j42 and cyp9j4,” have been deleted. And the new sentence has been rewritten in the line 267-282.

Author's response

The word “other” has been replaced by the word “tested” in the line 286.

Author's response

The word “of” has been replaced by the word “to” in the line 316 and 318.

Round 2

Reviewer 1 Report

The authors took into account the reviewers comments in their revision of their manuscript, but there are still a few things to modify throughout the text. After these minor revisions, I am favorable for this manuscript to be accepted for publication.

The different things to modify are:

The ace-1 gene should be in italics, by convention, as you did for all the P450 genes. This should be corrected lines 53, 123, and 335 (for the latter, it is not clear if it is the gene that is referred to) Please do not start sentences with "And", just remove it (lines 49, and 389) Line 125, you mention "the wild population"; are you referring to F0? if so, please replace by F0. Lines 220-222 is rather difficult to read. Please replace by: "Morever, with the LC25 propoxur treatment, a reduction of 6.3-16.2% was observed in egg-hatching, while the LC50 treatment resulted in a reduction of 5.3-15.5%." "Significantly" should be used instead of "significant" (lines 280, and 302) Also, lines 280-284, this sentence is long and difficult to read. Please replace by: "The level of P450 genes over-transcription was significantly higher with each propoxur treatment, compared to control. The LC25 propoxur treatment induced the over-transcription of 18 genes, out of the 32 tested P450 genes. With the LC50 treatment, 16 genes, out of the 32 tested P450 genes, were over-transcribed, compared to control. Moreover the LC25 and LC50 propoxur treatments significantly increased the expression of 14 genes under both treatments: [list the genes]"

Figure 1: The legend in the figure is missing. We don't know which group is which. Please add description either in the figure or in the legend. Figure 2: Please indicate it's a log-scale, by adding log graduations between the units 1, 0.1, 0.01, an so on. Tables: Your legend should read "The reported values are means ± SE. No need for brackets. Please modify accordingly in every table.